# A Lightweight System-On-Chip Based Cryptographic Core for Low-Cost Devices

**DOI:** 10.3390/s22083004

**Published:** 2022-04-14

**Authors:** Dennis Agyemanh Nana Gookyi, Kwangki Ryoo

**Affiliations:** Department of Information and Communication Engineering, Hanbat National University, Daejeon 34155, Korea; dennisgookyi@gmail.com

**Keywords:** IoT, low-cost device, lightweight cryptography, hardware architecture, SoC

## Abstract

The backbone of the Internet of things (IoT) platform consists of tiny low-cost devices that are continuously exchanging data. These devices are usually limited in terms of hardware footprint, memory capacity, and processing power. The devices are usually insecure because implementing standard cryptographic algorithms requires the use of a large hardware footprint which leads to an increase in the prices of devices. This study implements a System-on-Chip (SoC) based lightweight cryptographic core that consists of two encryption protocols, four authentication protocols, and a key generation/exchange protocol for ultra-low-cost devices. The hardware architectures use the concept of resource sharing to minimize the hardware area. The lightweight cryptographic SoC is tested by designing a desktop software application to serve as an interface to the hardware. The design is implemented using Verilog HDL and the 130 nm CMOS cell library is used for synthesis, which results in 33 k gate equivalents at a maximum clock frequency of 50 MHz.

## 1. Introduction

An Ericsson report [1] in 2016 forecasts that by the year 2022, there will be around 29 billion connected devices out of which peer-to-peer (P2P) networked IoT devices will take up about 62% of all the connected devices. The IoT platform is made up of low-cost sensing and data collecting devices connected over P2P networks [2,3,4]. The devices in the P2P networks are always connected and constantly transmitting and receiving data [5,6,7]. IoT devices generally consist of connected meters, wearables, consumer electronics, and connected vehicles. The Ericsson report also states that 20% of IoT devices are connected over unlicensed radios such as ZigBee, Wi-Fi, and Bluetooth. IoT devices are usually constrained in terms of hardware area, memory capacity, and processing power [8] making it difficult to implement standard cryptographic protocols to safeguard sensitive data. This problem could simply be solved if consumers were willing to pay more for secured devices. Many consumers would rather choose an inexpensive insecure device over an expensive secure device for the fact that they are unaware of the cost incurred to them when these devices are hacked.

The National Intelligence Council (NIC) of the United States in a 2008 report listed IoT as one of the disruptive civil technologies. This means that it is a technology with the potential to degrade the country’s power within the next fifteen years. It is therefore important to design algorithms that are power efficient for IoT devices. For that matter, it is important to study the smallest available device in an IoT platform. One of the smallest devices used in IoT platforms is the Radio Frequency Identification (RFID) tag. A popular tag is the Electronic Product Code (EPC), which is compliant with the International Organization for Standardization and the International Electrotechnical Commission (ISO/IEC) 18000-6C RFID air standard. The EPC Class-1 Generation-2 (EPC-C1G2) is an RFID tag used mostly for item identification in the Ultra-High Frequency (UHF) band. This particular tag was designed to be low cost and therefore has a lot of limitations such as low hardware footprints, low memory, and low power [9]. Some of the specifications and limitations of EPC-C1G2 tags are shown in Table 1. From the table, the tag has a maximum gate count of 10,000 out of which less than 2000 gates are allocated for security protocols. This limitation of low-cost devices such as RFID tags calls for security algorithms that are lightweight with moderate security levels.

There are a number of lightweight security protocols for key exchange, authentication, and encryption for low-cost Internet of thing (IoT) edge devices. Many types of research focus on optimizing individual protocols on specific hardware platforms. The focus of this work is therefore to implement a lightweight cryptographic System-on-Chip (SoC) that combines several lightweight primitives. The lightweight cryptographic SoC has the capability of generating shared keys, authenticating the shared keys, and encrypting/decrypting data for secured communication. This is done to investigate the hardware resource consumption and the feasibility of implementing a complete cryptographic SoC for low-cost IoT devices. The SoC-based cryptographic system consists of two lightweight encryption/decryption protocols (PRESENT [10] and NEW [11]), four Hopper-Blum type lightweight authentication protocols (HB [12], HB+ [13], HBMP [14], HBMP+ [15]), and a recently proposed lightweight key exchange protocol [16].

The contributions of the study are as follows:This work starts by quantifying in terms of money the cost to individuals when inexpensive low-cost P2P networked devices are hacked. This is done to bring awareness to the fact that end users suffer a monetary loss when they purchase insecure low-cost devices.A hardware architecture for unifying four HB-type lightweight authentication protocols is proposed and implemented.A hardware architecture for a new lightweight key exchange protocol is proposed and implemented.A hardware architecture for unifying two lightweight encryption/decryption protocols is proposed and implemented.Ten RISC-V processor cores are synthesized and from the synthesis reports, an ideal processor is selected for this work.An SoC-based lightweight hardware cryptographic system that consists of a key sharing protocol, four lightweight authentication protocols, and two lightweight encryption/decryption protocols is proposed and implemented.A desktop software application is designed to serve as an interface to verify the functionality of the hardware cryptographic SoC on a Field Programmable Gate Array (FPGA) board.

The rest of this work is structured as follows: Section 2 discusses some related works, Section 3 discusses the cost to consumers when insecure devices are hacked, Section 4 gives a summary of all the protocols in the lightweight cryptographic SoC, Section 5 gives an overall hardware architecture of the SoC, Section 6 talks about the implementation results and testing of the SoC, and the conclusion and future works are discussed in Section 7.

## 2. Related Works

Lightweight cryptography has recently received a lot of attention due to the constrained nature of IoT end devices. A number of lightweight algorithms and their hardware implementations have been proposed for key sharing, authentication, and encryption/decryption. The proposals are usually standalone hardware architectures that optimize an individual lightweight cryptographic algorithm. Most lightweight cryptographic SoCs are usually designed by companies and therefore the core architecture and performance metrics are not publicly available. It is impossible to find a lightweight cryptographic SoC that implements the same algorithms as this work. Nonetheless, this section will discuss some research that seeks to achieve the same objectives as this study.

Due to the need for secure communication between an implantable medical device and external users, the authors in [17] proposed a custom security Application-Specific Instruction Set Processor (ASIP) known as Smart-Implant Security Core (SISC). The SISC design is capable of key management, message authentication, message integrity, and confidentiality using symmetric cipher encryption. The work made use of an in-order RISC architecture processor with a five-stage pipeline. The main focus of the work was on energy consumption and therefore the individual algorithms or protocols for achieving the secured communication were not addressed. The SISC architecture was synthesized using a 90 nm CMOS cell library and utilized over 250,000 gates. An issue with the SISC architecture is that detailed hardware designs of the individual protocols are not provided for analysis and the use of 250,000 gates is a deal-breaker when it comes to very constrained devices such as the RFID tag.

To provide high flexibility when integrating cryptographic cores and processors in an embedded system, the authors in [18] presented a complete SoC embedded into Actel CoreMP7 FPGA. Actel CoreMP7 is the first ARM7 architecture that has been optimized for usage in FPGAs with no license required. The authors demonstrated the performance of the Actel CoreMP7 by integrating it with security primitives such as Elliptic Curve Cryptography (ECC), Secure Hash Algorithm (SHA), Advanced Encryption Standard (AES), and a True Random Number Generator (TRNG). Only the processor core and the TRNG were implemented in hardware while the rest of the cryptographic protocols were implemented as soft cores. The total gates consumed by the hardware implementation were 11,7620 at 12.5 Mhz clock frequency while the total bytes consumed by the soft core were estimated at 37,932. The issue with this work is that 11,750 gates for implementing just a TRNG and a processor core is not ideal for constrained devices. In addition, a 12.5 Mhz frequency is too low for a high throughput, low-cost application.

To enable end-to-end security in the IoT, authors in [19] presented the first hardware architecture for Datagram Transport Layer Security (DTLS), known as DTLS Engine (DE). The DE architecture is composed of a 2 KB RAM, 128-bit AES cipher, 256-bit SHA protocol, ECC primitives, and a 32-bit RISC-V processor core. The DE hardware architecture was synthesized using a 65 nm CMOS cell library which resulted in 149,000 gates and 6.75 KB SRAM at a maximum frequency of 16 MHz. The synthesis results indicate that the DE hardware architecture cannot be implemented in low-cost devices with constrained hardware gate requirements.

The SoC security architectures presented in this section are all designed for high-cost devices with no hardware gates constraints. This article, therefore, implements an SoC security architecture tailored toward very constrained devices. The security algorithms for this work which include a key exchange algorithm, authentication algorithms, and encryption/decryption algorithms are carefully selected to meet the lightweight cryptographic resource requirements discussed in Section 1.

### Features of the Proposed Hardware Architectures

For lightweight authentication, the four protocols (HB, HB+, HBMP, and HBMP+) are combined into a single module to authenticate 128-bit shared keys. The features of the proposed design, as opposed to conventional approaches, are listed in Table 2.

The lightweight key exchange unit generates 128-bit keys using 64-bit secret parameters. The features of the proposed design, as opposed to conventional approaches, are listed in Table 3.

The lightweight encryption/decryption unit consists of two ciphers (PRESENT and NEW) designed to use a single core. The features of the proposed design, as opposed to conventional approaches, are listed in Table 4.

The hardware architectures of the security protocols in the SoC architecture include a lightweight key exchange unit, a unified architecture of the four authentication algorithms, and a unified architecture of the two encryption algorithms. The features of the proposed design, as opposed to conventional approaches, are listed in Table 5.

## 3. Consumer Cost of an Insecure IoT Device

The big question is: how do you convince consumers and governments to invest in secured P2P networked IoT devices? The answer to this question is to devise a metric to calculate the cost when their devices are hacked. To do this (quantifying the cost of an insecure IoT device to consumers), this work makes use of a test module that was proposed by Kim et al. [20]. For the sake of research, they infected IP cameras with malware and measured the additional power and bandwidth consumption during the duration of the hack. The purpose of the malware is to infect one insecure device and scan for other devices resulting in a Distributed Denial of Service (DDOS) attack. Based on the results of the additional bandwidth and energy consumption, a calculator was developed to estimate the cost incurred by consumers when their devices are used in DDOS attacks. For the purpose of our case study, we use Ghana which is nearly a middle-income country with a fairly developed IoT industry that is now focusing on implementing smart metering systems [21]. We use the cost metrics to estimate the bandwidth and energy consumption on the economy of the country, as shown in Figure 1.

The average cost of electricity and bandwidth in Ghana is estimated at USD 0.32/kWh and USD 0.88/GB, respectively. The assumption is made that 24,000 insecure IoT devices are hacked and used in a DDOS attack for one hour. The combined estimated cost of bandwidth and electricity was found to be around USD 164,800 in just one hour of the attack. For a middle-income country, this could cause a lot of damage to the economy. This implies that IoT devices have to be designed with security as a priority.

## 4. Summary of Implemented Lightweight Cryptographic Algorithms

Designing lightweight cryptographic protocols is much needed in this era. These protocols usually trade hardware areas for medium security and throughput. Medium security protocols are enough to safeguard data on constrained devices because the data are continuously changing, making it difficult for adversaries to mount ciphertext-only attacks. The lightweight cryptographic core implemented consists of a lightweight key exchange protocol, four lightweight authentication protocols, and two lightweight encryption/decryption protocols. This section discusses only the algorithm flow of these protocols and for that matter, interested readers can refer to the references for more detailed descriptions.

### 4.1. Lightweight Authentication Protocols

Hopper and Blum developed the earliest form of a lightweight authentication protocol. The protocol is now officially known as the HB [12] authentication protocol. The HB protocol was designed to be very simple and to be used by humans without any form of computing aid. The first protocol has gone through a lot of analysis for its security and a number of different lightweight authentication protocols have been proposed. The security requirements of the HB-type protocols have been well defined but their real-world hardware implementations have been left out. Without real-world hardware implementation of the authentication protocols, it is difficult to determine if they are suitable for low-cost devices such as RFID tags. Four HB-type lightweight authentication protocols which include HB [12], HB+ [13], HBMP [14], and HBMP+ [15] are studied and implemented in this work. The lightweight authentication algorithms exploit the Learning Parity in the Presence of Noise (LPN) problem [22] to enable successful authentication. Due to the difficulty of solving the LPN problem, the HB-type lightweight authentication protocols rely on it for security.

The HB authentication protocol was proposed to be as simple as possible because it involved human authentication without computing devices. Figure 2a illustrates the authentication flow between the tag and the reader. Both the reader and the tag share a **k**-bit secret key **x**.

The HB+ authentication protocol is a variant of the HB protocol that was developed by Juels and Wess in 2005. It is also an improvement over the HB authentication protocol because it is stated to be resistant to active attacks. Figure 2b illustrates the authentication flow between the tag and the reader. Both the reader and the tag share two **k**-bit secret keys: **x** and **y**.

The HBMP authentication protocol is a variant of the HB+ protocol that was developed by Munilla and Peinado in 2007. It is stated to be well suited for low-cost hardware implementations such as the RFID tag. It is also an improvement over the HB+ authentication protocol because it is stated to be resistant to both active attacks and man-in-the-middle attacks. Figure 2c illustrates the authentication flow between the tag and the reader. The reader and the tag share two **k**-bit secret keys: **x** and **y**.

The HBMP+ authentication protocol is a variant of the HB protocol that was developed by Leng et al. in 2008. It is an enhanced version of the HBMP protocol that eliminates several vulnerabilities and keeps the simplicity of the original HB protocol intact. Figure 2d illustrates the authentication flow between the tag and the reader. Both the reader and the tag share a **k**-bit secret key: **x**.

### 4.2. Lightweight Key Exchange Protocol

The lightweight key exchange protocol was proposed by Ryoo et al. [16]. The lightweight key exchange protocol is based on the HB family of authentication protocols whose security is dependent on the LPN problem. To generate a shared key between two sides, both of them should have pre-shared parameters **x** (a **k**-bit key), **y** (a **k**-bit key), **ŋ** (the number of 1’s generated by a random bit generator), and a counter (**cnt**) variable. The top-level flow of the proposed key exchange protocol is shown in Figure 3. The key generation takes two simple steps. The first step is the generation of the shared key (**Vab**) by two entities while the second step is to ensure that both sides have the same shared parameters: **x** and **y**.

### 4.3. Lightweight Key Exchange Protocol

The security concerns of low-cost devices have led to the design of a number of lightweight encryption/decryption algorithms, some of which include PICCOLO [23], CLEFIA [24], PRESENT [10], and NEW [11]. This work focuses on implementing both PRESENT and NEW encryption/decryption algorithms. This is because of the use of minimal memory for storing their Substitution Boxes (S-BOX).

The PRESENT encryption/decryption algorithm is a Substitution-Permutation (SP) network architecture algorithm that encrypts/decrypts a block of 64-bit data in 31 rounds. It consists of two key lengths depending on the security level of the application it is used for. The two key lengths include an 80-bit key (for medium security applications with a low-area budget) and a 128-bit key (for high-security applications with a high-area budget). It implements a low-memory 4-bit input/output S-BOX and a low hardware area simple Permutation Box (P-BOX). The top-level pseudocode and algorithm flow for both the encryption and decryption routines are shown in Figure 4a,b, respectively.

The NEW encryption/decryption algorithm uses the Feistel architecture on a block of 64-bit data and a key length of 128 bit. The cipher uses just eight rounds for the whole encryption/decryption process. It implements the same S-BOX used by the PRESENT cipher. The P-BOX is made up of a one-stage omega permutation network structure. Figure 5a,b illustrate the encryption and decryption routines, respectively.

## 5. Proposed Hardware Architectures for Modules in the Lightweight Cryptographic SoC

Many lightweight cryptographic protocols proposed for low-cost P2P network devices, which include [25,26,27], do not come with hardware implementations. This makes it difficult to ascertain their suitability for extremely constrained devices. In this section, hardware architectures are proposed for the key exchange protocol, the four authentication protocols (HB, HB+, HBMP, and HBMP+), and the two encryption/decryption protocols (PRESENT and NEW). The hardware architectures are combined into a single lightweight cryptographic SoC with the ability to generate shared keys, authenticate the two entities wishing to communicate, and encrypt/decrypt data for secured communication. The functionality of the lightweight cryptographic core is extended by using a synthesizable RISV-V processor for controlling, scheduling, and reconfiguring the protocols. A Memory-Mapped Interface serves as a bridge between the processor core and the cryptographic engine. Peripherals such as Universal Asynchronous Receiver Transmitter (UART), LEDs, and seven-segment display are used for monitoring control signals from both the cryptographic core and the processor core. 

### 5.1. Proposed Hardware Architecture for Unifying the Lightweight Authentication Algorithms

The proposed hardware architecture that combines HB, HB+, HBMP, and HBMP+ lightweight authentication protocols into a single module is shown in Figure 6. The figure is a simplified structure that can be implemented at both the RFID reader and the tag side. This core is capable of authenticating a reader and a tag using 64-bit shared parameters. For simplicity, the figure shows only the datapath. The modules in the datapath include a rotation unit (for key rotation), a PRNG unit (for generating random numbers), a PRBG unit (for generating random bits), a dot product unit (for computing the dot product of the key and random numbers), a comparator unit, a key generation unit, an authentication unit, and four registers. The PRNG unit and PRBG unit are designed using the Linear Feedback Shift Register (LFSR) method. The dot product unit is designed using AND and XOR gates only. The key generation unit and the comparator unit are used by only HBMP and HBMP+ protocols. The Authentication unit is used by only the RFID reader to authenticate the tag. The control unit (not shown in the figure) uses a Finite State Machine (FSM) to generate signals for the modules in the datapath together with select signals for the multiplexors.

#### 5.1.1. PRNG and PRBG Units

AKARI1 [28] is one of the best-known PRNG for constrained devices. It uses about 1980 gates for implementation. The downside to this algorithm is that it uses about 66 clock cycles to generate one random number. This cannot be overlooked when the total authentication time is limited to a few thousand clock cycles [8].

The random numbers and bits for this work are generated using the LFSR method. One of the reasons for choosing the LFSR method is that it is implemented using only a number of flip-flops arranged sequentially where the output of one flip-flop feeds into the input of another and an XOR gate. The output of the XOR gate serves as the feedback signal and is fed into the first flip-flop in the sequence. For the PRNG unit (Pseudorandom Number Generator), a 64-bit random number is generated by concatenating four 16-bit LFSRs. It takes 16 clock cycles to produce the first random number. For the PRBG unit, a 4-bit LFSR is used to generate the random bits. It takes only about four cycles to produce the first random bit.

#### 5.1.2. Dot Product Unit

The scaler dot product is used by all HB-type authentication protocols to compute the product of random numbers and secret keys. The dot product unit can be implemented in a number of different ways that typically provide a trade-off between hardware area and latency. With 64-bit inputs, the dot product unit implemented utilizes 32 AND (&) gates and 32 XOR (ⴲ) gates for its computation of Equation (1). Using this method, the dot product of two 64-bit variables is computed in one clock cycle leading to fast computation at the expense of the hardware area. The hardware architecture for this method is shown in Figure 7.
dot product = ⴲ(input1[63:0] & input2[63:0])(1)

#### 5.1.3. Key Generation Unit

HBMP uses a key generation unit to generate 64-bit keys for every round, unlike HB and HB+. The key generation unit inputs include 64-bit **key1** and the Most Significant Bit (MSB) of **key2**. To generate a new key (**roundkey**), **key1** is rotated to the left by 1 bit if the MSB of **key2** is a 1. The key, however, remains unchanged if the MSB of **key2** is 0. The key generation is illustrated in Figure 8a.

The original HBMP+ proposal did not include an algorithm to generate round keys. This work, therefore, proposes a simple one-way hash function to generate unique keys for each round of the HBMP+ protocol. The proposed architecture of the one-way function and its pseudocode is shown in Figure 8b. The inputs to the function are the 64-bit **key** and the 64-bit **reader_ran_num**. From the figure, **A**, **B**, **C**, **D**, and **E** are initialized with **reader_ran_num** [31:0], **key** [31:0], **reader_ran_num** [63:32], **key** [63:32], and a 31-bit hexadecimal value (**9E3779B7**), respectively. **A** and **B** are left rotated by 5 bits and stored in **a** and **b,** respectively. The variable **f** is computed using AND (&), OR (|), and NOT (~) gates, as shown in Equation (2).

The key generation unit hardware architecture for the unified protocols is shown in Figure 8c. The figure combines the key generation of the HBMP and the HBMP+ protocols described above.
f = (B & D)|(~B & E)|B (2)

#### 5.1.4. Authentication Unit

The authentication unit is a simple logic that indicates whether the tag is valid or not after the whole authentication process. It takes as its inputs two 6-bit variables: **noise_cnt** and **cnt**. The **noise_cnt** and **cnt** values are passed through a comparator and if they are equivalent, the tag is deemed valid, otherwise the tag is deemed invalid. The comparator is designed using only AND gates as shown in Figure 9.

#### 5.1.5. Synthesis Results for the Proposed Unified Authentication Hardware Architecture

The proposed hardware architecture that unifies HB, HB+, HBMP, and HBMP+ was designed using Verilog HDL. Xilinx ISE 14.7 was used to synthesize the design using a Virtex4 XC4VLX80 FPGA device. The synthesis report produced the total number of slices, the total number of LUTs, and the maximum frequency of the design. The design utilized 786 slices, 1434 LUTs, and achieved a maximum frequency of 176.6 MHz. The results are shown in Table 6. The table also shows the results of the synthesis of the individual protocols reported in a previous work by the authors [29].

The synthesis results in [29] are for only the RFID tag side of the protocols and for that matter the tag together with the reader architecture will double the resources used, as shown in Table 6. Combining the individual protocol architectures given in [29] results in 1094 slices and 1910 LUTs. As compared to the proposed unified architecture, the number of slices is reduced by 28.15% while the number of LUTs is reduced by 24.92%.

### 5.2. Proposed Hardware Architecture for the Lightweight Key Exchange Protocol

The majority of lightweight security proposals do not come with real-world hardware architecture implementations. This makes it difficult to ascertain their feasibility when it comes to low-cost devices. This section describes the proposed hardware implementation of the lightweight key exchange protocol.

The proposed hardware architecture of the lightweight key exchange protocol is shown in Figure 10. The same hardware architecture can be implemented by all devices wishing to generate a 128-bit shared key. The architecture consists of both a datapath and a control unit. The inputs to the datapath include a 128-bit **seed** value, a 128-bit **key_x**, a 128-bit **key_y**, computed value **zb** from the communicating partner, 7-bit **noise_cnt**, and **noise_parameter** from the control unit. The outputs include a 128-bit random number (**a**), computed value **za**, a 128-bit generated shared key (**shared_key**), and a **valid** signal. The modules in the datapath include a PRNG unit, a PRBG unit, a COMPUTE1 unit, a COMPUTE2 unit, a VERIFICATION unit, an XOR gate, and a SHIFT REGISTER unit.

The PRNG unit and the PRBG unit are discussed in Section 5.1.1. The PRNG unit used here generates a 128-bit random number using a 128-bit **seed** from the seed register while the PRBG unit generates a random bit using a 4-bit seed from the **seed** register.

#### 5.2.1. COMPUTE1 and COMPUTE2 Units

The COMPUTE1 unit computes the output **za** by using as its inputs 128-bit value **key_x**, 128-bit random number **a**, and 1-bit random bit **va**. The **key_x** and **a** values are passed through a DOT PRODUCT. The DOT PRODUCT unit architecture is discussed in Section 5.1.2. The output from the DOT PRODUCT unit is XORed with the random noise bit (**va**) to produce the output **za**. The output value **za** is sent to the communicating partner and is used for the computation of the random bit generated. The hardware architecture of the COMPUTE1 unit is illustrated in Figure 11a.

The COMPUTE2 unit estimates the random bit (**vb**) generated by the communicating partner by using as its inputs 128-bit value **key_x**, 128-bit value **key_y**, 128-bit random number **a**, 128-bit random number from the partner (**b**), and 1-bit computed value from the communicating partner (**zb**). The **key_x** and **b** values are passed through a DOT PRODUCT unit while **key_y** and **a** are also passed through a second DOT PRODUCT unit. The outputs from the DOT PRODUCT units are XORed with the **zb** value which results in the computation of the **vb** value. The architecture of the COMPUTE2 unit is shown in Figure 11b.

#### 5.2.2. VERIFICATION Unit

The VERIFICATION unit is similar to the authentication unit discussed in Section 5.1.4. The VERIFICATION unit is a simple logic that indicates whether the shared key generated is valid or not after the whole key generation process. It takes as its inputs two 7-bit values—**noise_cnt** and **noise_parameter**—from the control unit. It also takes in **vb** from the COMPUTE2 unit. The **noise_cnt** and **noise_parameter** values are passed through a comparator and if they are equivalent, the shared key is deemed valid, otherwise it is deemed invalid. The comparator is designed using only AND gates. The hardware architecture for the VERIFICATION unit is shown in Figure 12.

#### 5.2.3. Synthesis Results for the Proposed Key Exchange Hardware Architecture

The hardware architecture of the proposed lightweight key exchange protocol was designed using Verilog HDL and synthesized using Xilinx ISE 14.7 software. The ISE software is embedded with the XPower Analyzer, which was used to estimate the power consumption of the design. The design was implemented on a Virtex4 FPGA device.

Since most lightweight key exchange protocols do not come with a real-world hardware implementation, the proposed design was compared with the compact version of the Rivest–Shamir–Adleman (RSA) [30] and the Elliptic-Curve Diffie–Hellman (ECDH) [31] public key protocols. The metrics for comparison includes the hardware area, maximum operating frequency, latency, and power consumption. The results of the comparisons are shown in Table 7.

From the table, the proposed design used 312 slices, achieved a maximum frequency of 432 MHz, generated a shared key in 76 µs, and consumed power of 33 mW. With regards to hardware area measured in slices, the proposed design achieves an area-saving of 93% and 95% as compared to ECDH and RSA, respectively. In terms of maximum achievable frequency measured in MHz, the proposed design is four times and three times higher than that of RSA and ECDH, respectively. With regards to latency, it takes 420 times and three times more time to generate a shared key using RSA and ECDH as compared to the proposed design. Finally, ECDH consumes 12 mW more power than that of the proposed design while the power consumption of RSA was not recorded.

### 5.3. Proposed Hardware Architecture for Unifying the Lightweight Encryption Protocols

This work studied two lightweight encryption/decryption protocols which include PRESENT cipher and NEW cipher. The two algorithms are used for securing data in very constrained devices. The algorithms use the same S-BOX for obscuring the relationship between the key and the ciphertext. Here, a hardware architecture for unifying the two lightweight ciphers is proposed and implemented.

Figure 13 illustrates the datapath of the proposed hardware architecture for unifying PRESENT cipher and NEW cipher. The architecture is used by the two algorithms for both the encryption and decryption routines. From the figure, the input consists of 64-bit wire INPUT DATA. This could be either the plaintext or the ciphertext since the architecture is designed for both. The output consists of the 64-bit wire OUTPUT DATA. The datapath uses three registers for storing intermediate data. Reg1_1 and Reg1_2 are two 32-bit registers for storing computed intermediate data. The values from the two registers are concatenated and stored in the Reg2 register. The Reg2 register is a 64-bit register that serves as the output. The architecture implements two 32-bit S-BOXes (SBOX1 and SBOX2) together with two 64-bit P-BOXes (PBOX1 and PBOX2). RK1, RK2, RK3, and RK4 represent 32-bit selected parts of the ciphers’ round keys. A total of four XORs gates are employed with a number of multiplexors for channeling data based on specific select signals. From the figure, the red short-dashed lines indicate the route of NEW encryption/decryption signals while the green long-dashed lines indicate the route of the PRESENT encryption/decryption signals. The black lines indicate the route common to both algorithms.

#### 5.3.1. Proposed Hardware Architecture for a Unified S-BOX

The PRESENT algorithm makes use of an S-BOX and an inverse S-BOX while the NEW algorithm uses only an S-BOX without the inverse S-BOX. The hardware architecture implements two 32-bit SBOXes. Figure 14 shows an illustration of one of the S-BOXes. The 32-bit input S-BOX was implemented by using eight 4-bit input/output S-BOXes. From the figure, both the S-BOX and inverse S-BOX are implemented in one core. The select signals **protocol** and **enc** are used to select the type of cipher and whether the operation is encryption or decryption, respectively. The 32-bit S-BOX is implemented to reduce latency and increase throughput at the expense of a slight increase in the hardware area.

#### 5.3.2. Proposed Hardware Architecture for a Unified S-BOX

The PRESENT algorithm P-BOX involves bit permutation while that of the NEW algorithm uses a one-stage omega permutation network that relies on the cipher key. The datapath instantiates two 64-bit P-BOXes. The architecture of the 64-bit unified P-BOX together with the pseudocode for both the PRESENT cipher and NEW cipher is shown in Figure 15. It takes 64-bit input and produces 64-bit output. The NEW algorithm only makes use of the 32-bit Least Significant Bit (LSB) while PRESENT makes use of the whole 64-bit. The select signals **protocol** and **enc** are used to choose between the algorithms and whether encryption or decryption is performed, respectively.

#### 5.3.3. Proposed Hardware Architecture for a Unified Key Schedule

The hardware architecture for unifying the key schedule algorithm for both PRESENT and NEW ciphers is illustrated in Figure 16. From the figure, the input consists of the 128-bit **KEY_IN** while the output is the 128-bit **KEY_OUT**. The architecture consists of two registers—**Reg1** and **Reg2**—for storing intermediate computation values. The **ROTATION** module rotates the input key by 25-bit, 61-bit, and 19-bit depending on the type of algorithm and the functionality (encryption or decryption) of the algorithm. The S-BOX is used by the PRESENT encryption/decryption algorithm. For the PRESENT cipher, the input key is rotated by 61-bit for encryption and 19-bit for decryption. Part of the rotated key is passed through the S-BOX while a part is XORed with the **COUNTER** value. For the NEW cipher, since the key schedule for both encryption and decryption algorithms are the same, the input key is just rotated by 25 bits. From the figure, the green long-dashed lines indicate the path for routing NEW encryption/decryption signals while the red short dash lines are for PRESENT encryption/decryption signals. The black continuous lines are for both algorithms.

#### 5.3.4. Synthesis Results for the Proposed Unified Encryption/Decryption Architecture

The proposed hardware architecture that unifies the PRESENT cipher and NEW cipher was designed using Verilog HDL. Xilinx ISE 14.7 was used to synthesize the design using a Virtex4 XC4VLX25 FPGA device. The synthesis report produced the total number of slices, the total number of flip-flops, the total number of LUTs, and the maximum frequency of the design. The design utilized 863 slices, 199 flip-flops, 1431 LUTs, and achieved a maximum frequency of 50 MHz. The results are shown in Table 8. The table also shows the results of the synthesis of the individual protocols by [10] (PRESENT algorithm) and [11] (NEW algorithm).

References [10,11] recorded the synthesis results for only the encryption algorithm of PRESENT and NEW, respectively. Here, an assumption is made that the decryption algorithm uses the same number of hardware resources as the encryption algorithm. When the hardware resource used by the individual PRESENT cipher and NEW cipher are combined, the total slices amount to 834, the total flip-flops amount to 794, and the total LUTs amount to 1520. As compared to the proposed unified encryption/decryption architecture, the number of flip-flops is reduced by 74.9% while the number of LUTs is also reduced by 5.9%. The number of slices, however, increased by just 3.3%.

### 5.4. Selecting an Ideal RISC-V Processor Core for a Lightweight Cryptographic SoC

In the development of IoT that sense, store, and transmit sensitive information, the hardware architecture should implement functionalities such as trusted Intellectual Property (IP) and secure communication. To achieve these functionalities, the best solution is to design the application using processor cores that implement open ISAs like RISC-V [32] rather than the standard ARM or x86-based processors. An open ISA gives a designer the advantage of implementing an architecture that is best for a specific application. This makes the designer innovative in implementing designs with the lowest power consumption. An additional advantage of using open ISAs is that designers can obtain processor source codes from vendors. With the source code available, a deep inspection can be done to prevent the insertion of malicious code by some vendors.

RISC-V is a relatively recent ISA that was originally designed for academic purposes. It has quickly become an open-source ISA for both research and industry implementations. The main goal for designing the RISC-V ISA was for it to be freely available for both academia (using small base integer ISA) and industry (using higher base integer ISA).

This section briefly examines ten synthesizable processors that implement the RISC-V ISA. Interested readers should see the references for detailed descriptions. The processors include Roa Logic RV12 [33], ORCA [34], SiFive E31 [35], SCR1 [36], Rocket [37], BOOM [38], MRISCV [39], PICORV32 [40], Shakti-E [41], and Hummingbird [42]. The Register Transfer Level (RTL) codes of the processors are downloaded from their respective repositories. The most important step in selecting a processor core for low-cost hardware devices is to evaluate its hardware resource consumption. This is done to make sure that the processor meets the resource limitation of the hardware device.

#### Synthesis Results for the Proposed Unified Encryption/Decryption Architecture

Vivado v2018.02 is used to synthesize the RISC-V processor cores to obtain the hardware resources used for the implementation of each core. Vivado is a Computer-Aided Design (CAD) software tool that automatically converts high-level description languages such as HDLs into gate level. The tool generates area, power, and performance estimate that are important in implementing a design in the real world. These estimates are usually very fast and less expensive to produce yet they accurately depict the real implementation of a design. It is important to note that processor benchmarking programs such as Dhrystone [43], Stanford [44], Paranoia [45], and EEMBC [46] are good for measuring and comparing the execution time of processors. However, in low-cost designs, low hardware area and low power cores are preferred over high-speed modules. Therefore, the first step in selecting a suitable synthesizable processor for low-cost devices is to measure the hardware resources used by the processors.

Figure 17 gives the compilation of all the results from synthesizing the ten processors using the XC7Z020 FPGA device. The results include the maximum attainable operational frequency (Freq), the total number of LUTs, the total number of FFs, the total number of BRAMs, and DSPs.

Over 80% of the SoC devices in the IoT market consist of components such as processors, buses, power management units, security protocols, and other peripherals. The processor core usually takes up less than 5% of the hardware resources. The number of LUTs utilized together with the power consumed are the most important metrics for FPGA designs. From the synthesis reports, PICORV32, ORCA, and MRISCV processors consumed 1.7%, 2.55%, and 3.56% of the available LUTs, respectively. In terms of area, the three processors are suitable for low-cost devices. In terms of power consumption, MRISCV and ORCA consumed two times and five times that of PICORV32, respectively. This implies that PICORV32 is much more suitable for low-area and low-power applications than the other processors. It is evident that the BOOM processor core uses the most hardware resources and consumes the most dynamic power as compared to the other processor cores and therefore is not suitable for low-cost devices. PICORV32 is, therefore, the chosen RISC-V processor core used for this work.

## 6. Overall Hardware Architecture for a Lightweight Cryptographic SoC

The overall hardware architecture of the proposed lightweight cryptographic SoC as shown in Figure 18 consists of a lightweight cryptographic core, PICORV32 RISC-V synthesizable processor, on-chip memory, UART, LEDs, and a seven-segment display.

The lightweight cryptographic core consists of a lightweight key exchange unit for shared key generation, a lightweight authentication unit for the shared key authentication, and a lightweight encryption/decryption unit to encrypt/decrypt data for secured communication.

The lightweight key exchange unit generates 128-bit keys using 64-bit secret parameters. The security of the protocol is based on the LPN problem. The hardware architecture of the protocol consists of a PRNG unit, a PRBG unit, an XOR gate, and a shift register. 

The four authentication protocols (HB, HB+, HBMP, and HBMP+) are combined into a single module. The module can be configured as either the authenticator or the “authenticate”. The core consists of internal modules such as the pseudorandom number and bit generators (which are designed with linear feedback shift registers); a dot product unit computes the dot product of the random numbers and the shared keys. Other modules include a comparator unit, an exclusive-or unit, a one-way hash function unit, and a key rotation unit.

The two ciphers (PRESENT and NEW) are designed to use a single core. The core can be configured as either an encryption core or a decryption core. The input to the core is the authenticated shared key (128-bit for NEW and 80-bit for PRESENT) and either a 64-bit plaintext or ciphertext. The output to the core, depending on the configuration, could be either a 64-bit ciphertext or the plaintext. The core implements a 32-bit module for both the S-BOX and the P-BOX. This is done to reduce the latency of the encryption/decryption process. The core also implements one key generation logic for both algorithms.

### 6.1. Synthesis and Simulation Results for the Proposed Lightweight Cryptographic SoC

The SoC-based lightweight cryptographic system was designed using Verilog HDL and synthesized using the 130 nm CMOS cell library. The synthesis report produced the total gate count and the maximum frequency. The design used a total of 33 k logic gates and achieved a maximum frequency of 50 MHz, as shown in Table 9. It is challenging to compare this lightweight cryptographic SoC to other systems because different algorithms are usually used in different systems. However, the results from the proposed design are compared to a DTLS engine proposed by Utsav et al. [19]. The algorithms used by [19] include Elliptic Curve Digital Signature Algorithm (ECDSA), Secure Hash Algorithm (SHA-258), and AES-128. They used a processor that implements the RISC-V RV32I ISA and synthesized their design using a 65 nm CMOS cell library. The design consumed a total of 149 k logic gates and achieved a maximum frequency of 16 MHz. The results are tabulated in Table 9.

In terms of logic gates, Reference [19] used 76.5% more gates as compared to the proposed design. The proposed design frequency is a little over three times more than that of [19]. Lastly, the proposed design is 15 times more efficient (dividing the frequency by the logic gates) than that of [19].

A number of factors account for the efficiency of the proposed design as compared to the DTLS engine. The first reason is that the proposed design selected purely lightweight cryptographic algorithms while the DTLS engine selected compact versions of standard cryptographic algorithms that are not suitable for low-cost IoT devices. Moreover, the hardware architectures of the proposed design were made to share resources to reduce area while that of the DTLS engine implemented standalone algorithms. Lastly, the RISC-V processor core selected for the DTLS engine consumed over 20% of the gate count while that for this work consumed less than 10%. This is because a lot of effort went into selecting a low-cost processor by synthesizing a number of RISC-V processors. In conclusion, the proposed architecture in this work performed better in terms of gate count and frequency which makes it suitable for low-cost IoT devices.

The functional simulation flow of the proposed lightweight cryptographic SoC involves a software part and a hardware part. The software part involves the generation of machine code to be loaded into the memory of the hardware part for the processor to execute. The hardware part consists of the integration of the processor core with the cryptographic SoC and memory. Figure 19 shows the simulation results in terms of time and data exchanged between two entities. The simulation frequency is set to 100 MHz. The simulation involves four phases. Phase 1 involves processor initialization and protocol selection. Phase 2 involves shared key generation. Phase 3 involves shared key authentication where HB used the least time and least data exchanged. Phase 4 involves data encryption/decryption where the NEW cipher uses the least time, and both use the same number of data exchanged.

### 6.2. FPGA Board Verification of the Proposed Lightweight Cryptographic SoC

The SoC-based lightweight cryptographic system was verified on an FPGA board using a desktop application. For verification on an FPGA board, three main design flows are used. This includes the FPGA hardware design flow, the processor software code design flow, and a desktop API design flow. The design flows are illustrated in Figure 20.

The FPGA design flow starts with the design entry phase where all the algorithms and functionalities of the design are coded using Verilog HDL. The Verilog codes are simulated to meet the specification during the functional simulation phase. The codes are then synthesized to generate the netlist file. The netlist file is implemented by passing it through Translate, Map, Place, and Route phases. The output executable file from the processor software code design flow is loaded onto the memory of the hardware design before the final **bit** file is loaded onto the FPGA board. The FPGA board used for verification is the HB-SOC-IPD test board developed by Hanback Electronics. The board is equipped with the Xilinx Virtex4 XC4VLX80 FPGA device.

The processor software code design flow is used to write instructions to be executed by the processor hardware core. The flow starts with the Design Entry phase where programming languages such as C and ASM are used to generate the processor source files. The next phase in the design flow is the Generate Object Files phase where the source codes are compiled and their object files are generated. RISC-V ISA comes with its own toolchain and compiler for the generation of the object files. The compiler used is a 32-bit version known as the **riscv32-unknown-elf-gcc**. The last phase in the design flow is the Generate Executable File (**ELF**) phase. The object file and the linker files are combined to generate the **elf** file. The **elf** file contains the machine code for the processor. The **elf** file is loaded onto the memory of the hardware design before the final bit file is loaded onto the FPGA test board.

The desktop API design flow is used to implement an interface between the processor source code and the FPGA hardware. The flow starts with the Design Entry phase where the Microsoft Visual Studio editor is used to design source files using the C++ programming language. The Microsoft Visual Studio Integrated Development Environment (IDE) consists of the Visual Studio C++ compiler which is used to generate the object file in the Generate Object Files phase. The Generate Executable File phase converts the object file into an executable file that can be run to generate the software interface. The desktop application interface is designed to send data and control signals to the hardware module on the FPGA test board.

#### FPGA Board Verification Setup and Results Display

Figure 21 shows the setup used to test the hardware architecture of the proposed lightweight cryptographic SoC. The setup consists of a test board (HBE-SOC-IPD) that is equipped with an FPGA device that implements the proposed SoC, UART for transfer of data to and from the test board, LEDs, and segment display for signal monitoring. The setup also consists of a computer with a desktop application that serves as an interface to the FPGA test board. The desktop application Graphical User Interface (GUI) was designed using Microsoft Visual Studio software.

The desktop GUI that serves as an interface to the hardware consist of two panels. The first panel is the LIGHTWEIGHT KEY EXCHANGE PANEL while the second panel is the LIGHTWEIGHT KEY AUTHENTICATION AND ENCRYPTION PANEL.

The LIGHTWEIGHT KEY EXCHANGE PANEL is shown in Figure 22. From the figure, the user first connects to the appropriate communication port using the **UART SETUP** buttons. The user then clicks to display the seed and the noise parameter used at both **PARTY A** and **PARTY B** sides. The 256-bit shared key parameter is then entered which is divided into 128-bit **Key_X** and 128-bit **Key_Y**. The shared key generation parameters are sent to the key generation algorithm. The **Start Shared Key Generation** button is then clicked to begin the key generation process at **PARTY A** and **PARTY B**. The **Display Generated Key** buttons at both **PARTY A** and **PARTY B** are clicked to show the shared key generated by the lightweight key exchange protocol. The **Key Validity** buttons at both **PARTY A** and **PARTY B** are clicked to determine whether or not the generated shared keys are equivalent. From the figure, the **Key Validity** indicates that the shared keys generated at both **PARTY A** and **PARTY B** match.

The LIGHTWEIGHT KEY AUTHENTICATION AND ENCRYPTION PANEL is shown in Figure 23. From the figure, the user chooses the type of authentication from the **HB**, **HB+**, **HBMP**, and **HBMP+** buttons. This information is sent to the unified lightweight authentication hardware architecture. To determine whether or not the same keys are used at both **PARTY A** and **PARTY B**, the **Authentication Status** button is clicked. As shown in the figure, the keys match. The next step is the encryption of data using the authenticated key. The user selects the encryption algorithm to use by clicking either the **NEW** or **PRESENT** button. The information is sent to the unified encryption/decryption hardware architecture. The 64-bit plaintext is entered and sent to the hardware architecture. The **Start Encryption** button is clicked to begin the encryption process. The **Display Encrypted Data** button is clicked to show the 64-bit encrypted ciphertext. The user then clicks on the **Start Decryption** button to begin the decryption processes, after which the **Display Decrypted Data** button is clicked to show the decrypted data. From the figure, it can be observed that the decrypted data is equivalent to the plaintext value.

## 7. Conclusions and Future Direction

The backbone of the IoT revolution is made of tiny, networked devices that gather/sense data, process the data, and transmit the data over wireless mediums. The devices deal with very sensitive information ranging from security codes to the human heartbeat. It is therefore vital to secure the information stored in these tiny devices as much as possible. These devices are extremely low cost with severe constraints such as computational complexity, memory capacity, hardware gate counts, and power consumption. For that matter, the implementation of traditional cryptographic algorithms such as AES, RSA, and ECDH and hash algorithms is deemed infeasible. Implementing cryptographic algorithms in low-cost devices has created the branch of lightweight cryptography. Lightweight cryptographic algorithms are usually designed to use minimal hardware resources with medium security levels. 

This work starts by examining research works that measure resource utilization when insecure IoT devices are infected with malware. In [20], insecure IoT devices are simulated to participate in a DDOS attack. The energy consumption and bandwidth utilization before and during the attack are measured. The research in [20] was used by this work to develop a cost calculator for insecure IoT devices. The cost calculator was used to estimate the effect on the economy of middle-income countries when insecure devices are infected with malware. From this research, the energy and bandwidth consumption of an infected insecure IoT device could cripple most middle-income economies and for that matter, IoT devices should be designed with security as a priority.

To authenticate low-cost devices, this work examined four lightweight authentication protocols. The protocols are collectively termed HB-type authentication protocols which include HB, HB+, HBMP, and HBMP+. The securities of the HB-type protocols are well established but there are no real-world hardware implementations of the protocols. A hardware architecture for unifying the four protocols was proposed and implemented. This architecture resorts to resource sharing to reduce the hardware area. The synthesis results indicate that the unified hardware architecture is suitable for low-cost devices.

To generate shared keys for low-cost devices, this work examined a recently proposed lightweight key exchange protocol. The security of the key exchange protocol is the same as that of the HB-type authentication protocols. A hardware architecture is designed and implemented for the key exchange protocol. This consists of PRNG, PRBG, DOT product, XOR gate, and a shift register. The synthesis results of the hardware architecture confirm that the key exchange protocol is suitable for extremely constrained hardware devices.

To protect data stored in low-cost devices, this work examined two lightweight encryption/decryption protocols. The protocols include PRESENT cipher and NEW cipher. Both ciphers use the same 4-bit input/output S-BOX. A hardware architecture for unifying the two ciphers is proposed and implemented. The hardware architecture consists of a unified module for the S-BOX, P-BOX, and key scheduling algorithm. The hardware architecture shares resources to minimize the hardware area.

To implement a lightweight cryptographic SoC, a synthesizable processor core is vital. This work, therefore, set out to find a suitable lightweight processor core. A total of ten RISC-V synthesizable processor cores are examined. The cores are synthesized to extract the maximum frequency, LUT utilization, flip-flop utilization, LUTRAM utilization, BRAM utilization, DSP utilization, and power consumption reports. From the reports, the PICORV32 processor core synthesis results are ideal for low-cost devices and therefore were selected for this work.

The three hardware architectures which include the unified authentication, key exchange, and unified encryption/decryption together with the PICORV32 synthesizable processor are combined to form an SoC-based lightweight cryptographic system. The overall hardware architecture of the lightweight cryptographic SoC is synthesized using a 130 nm CMOS cell library which reports 33 k logic gates at 50 MHz maximum clock frequency. This is a suitable result for low-cost IoT devices. The system is verified on an FPGA board by designing a desktop application to serve as an interface to the hardware. Data and control signals are sent from the desktop application to the lightweight security module on the FPGA board for key generation, authentication, and encryption/decryption.

In summary, this work implemented a lightweight cryptographic SoC capable of key generation, authentication, and encryption/decryption. This is vital because IoT end devices are usually designed without security protocols as a priority. This is mainly due to the constrained nature of the IoT end devices. The synthesis results of the lightweight cryptographic SoC indicate that it can be implemented in constrained low-cost devices. 

The security of the proposed lightweight cryptographic SoC depends on secret keys that are used during the key generation and authentication process. The key generation and authentication processes use pseudorandom generators that rely on secret keys to generate random numbers and bits. A potential attack on the system will be the retrieval of these secret keys. This can be done by brute force or reverse-engineering the netlist file to recover the secret keys. This potential attack can however be averted by implementing Physically Unclonable Functions (PUFs) or true random number generators for generating random numbers and bits during the key generation and authentication processes.

The limitation of the proposed design is that thorough security analyses such as side-channel analysis have not yet been carried out on the design. This is left for future work. Future work will also extend the algorithms to include lightweight hash functions and message authentication codes for lightweight digital signature generation.

## Figures and Tables

**Figure 1 sensors-22-03004-f001:**
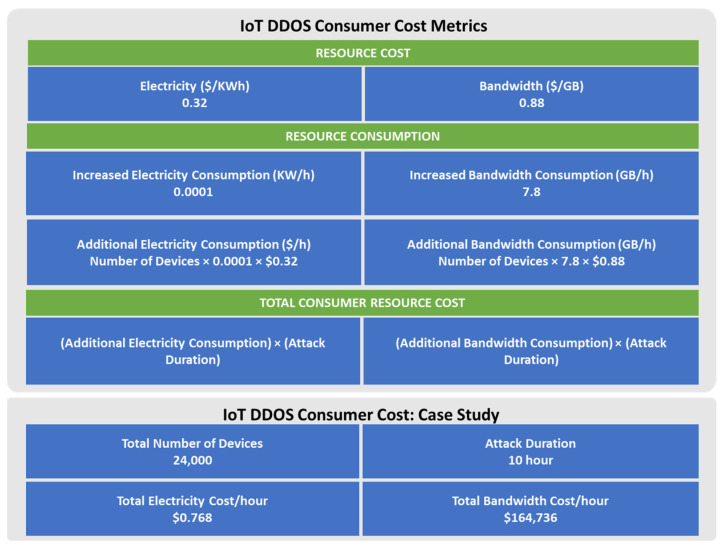
Quantifying the cost of insecure IoT devices.

**Figure 2 sensors-22-03004-f002:**
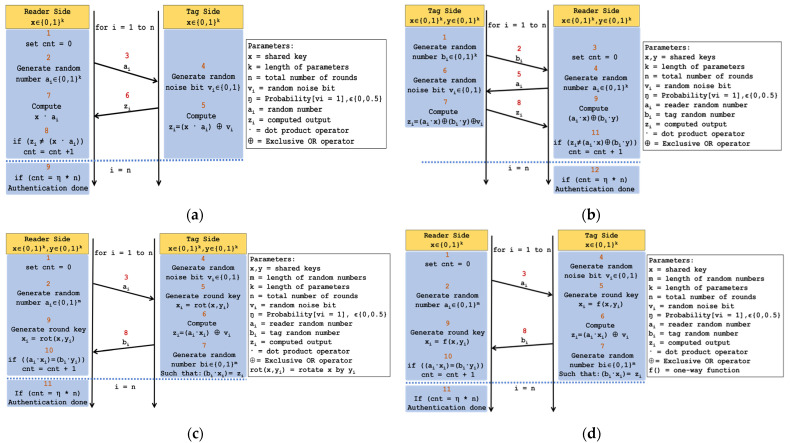
Algorithm flow of HB type lightweight authentication protocols: (**a**) HB protocol; (**b**) HB+ protocol; (**c**) HBMP protocol; (**d**) HBMP+ protocol.

**Figure 3 sensors-22-03004-f003:**
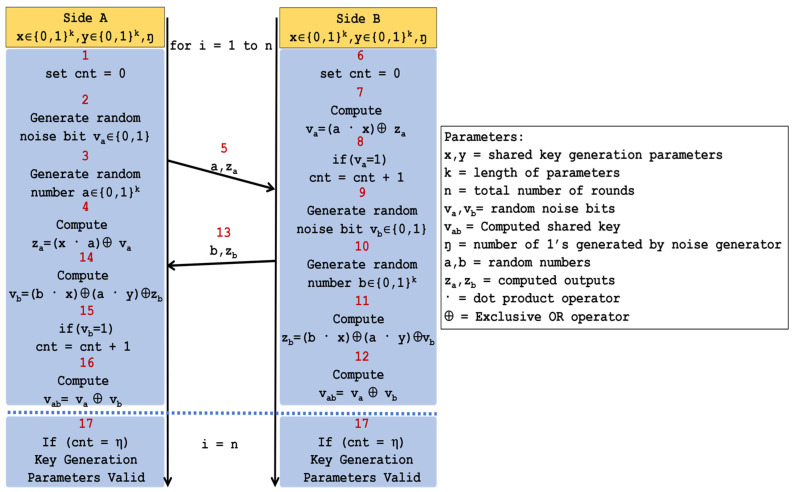
Algorithm flow of the lightweight key exchange protocol.

**Figure 4 sensors-22-03004-f004:**
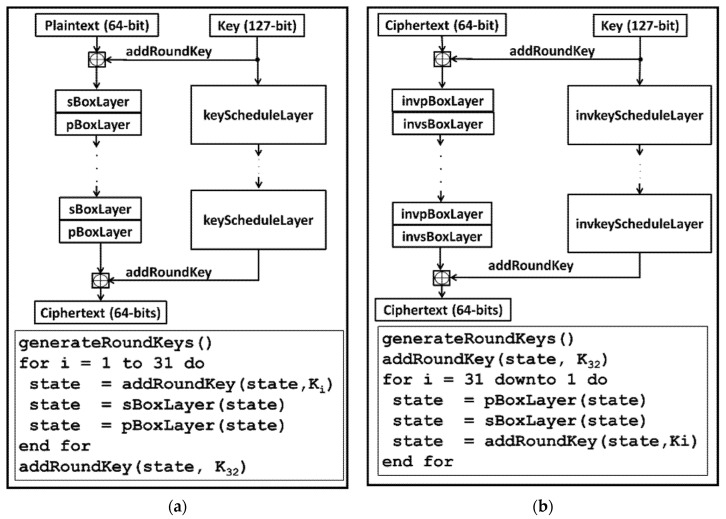
Top level description of PRESENT algorithm: (**a**) PRESENT encryption routine; (**b**) PRESENT decryption routine.

**Figure 5 sensors-22-03004-f005:**
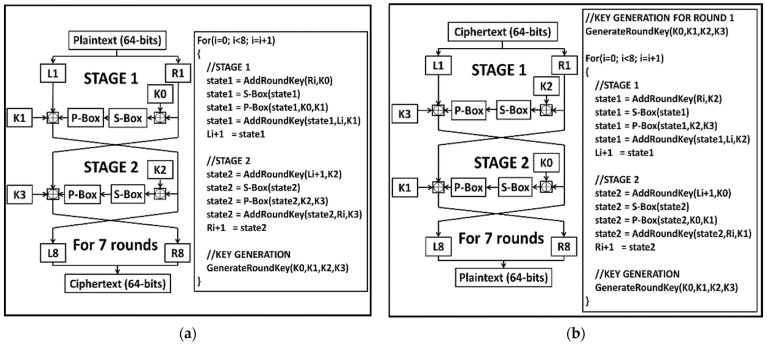
Top level description of NEW algorithm: (**a**) NEW encryption routine; (**b**) NEW decryption routine.

**Figure 6 sensors-22-03004-f006:**
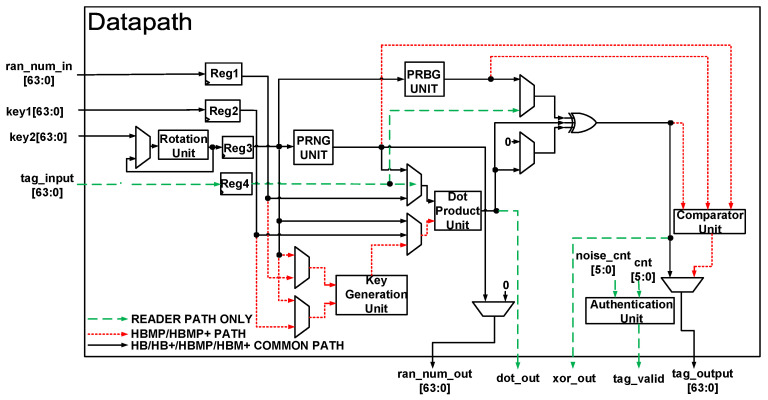
Proposed datapath for unifying the lightweight authentication protocols.

**Figure 7 sensors-22-03004-f007:**
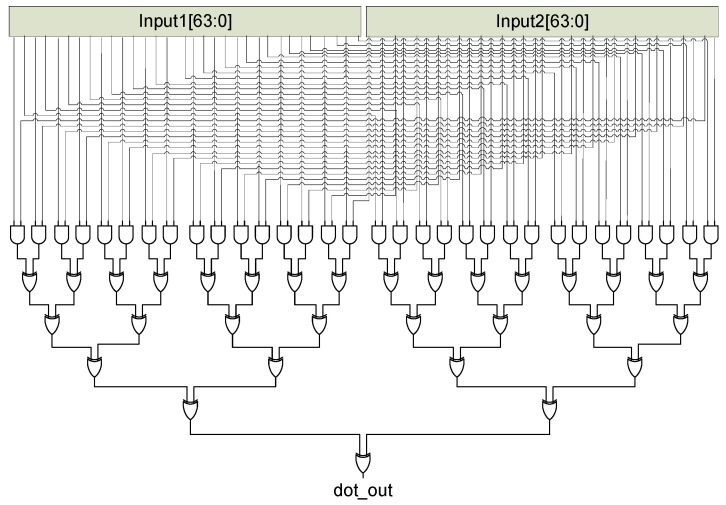
Proposed high latency dot product unit.

**Figure 8 sensors-22-03004-f008:**
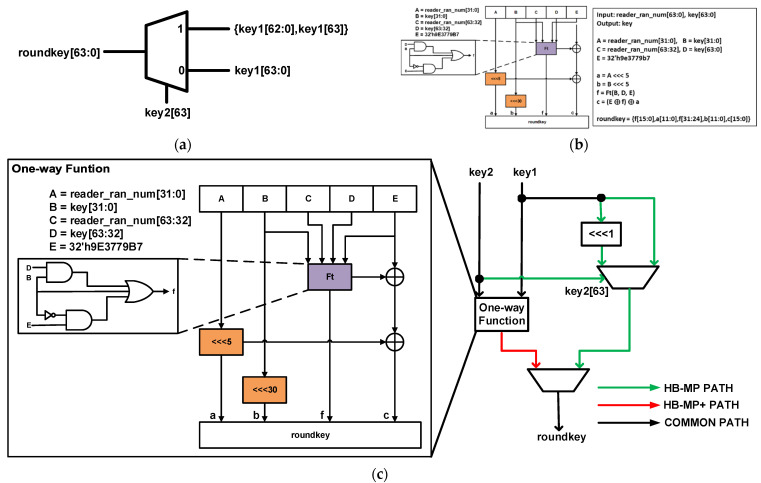
Proposed key generation unit: (**a**) HBMP round key generation; (**b**) HBMP+ round key generation; (**c**) Unified key generation for both HBMP and HBMP+.

**Figure 9 sensors-22-03004-f009:**
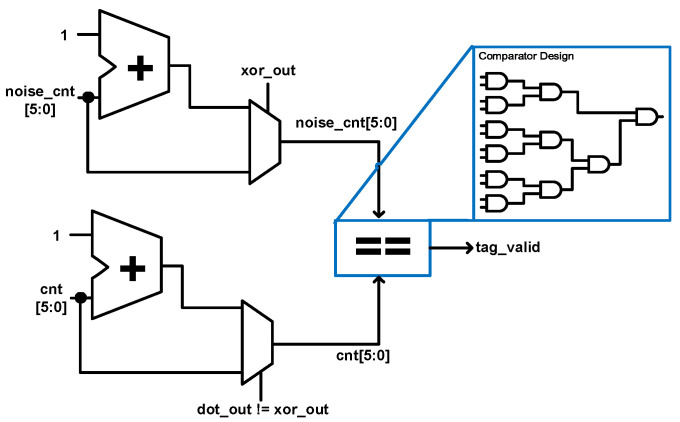
Proposed unified authentication unit.

**Figure 10 sensors-22-03004-f010:**
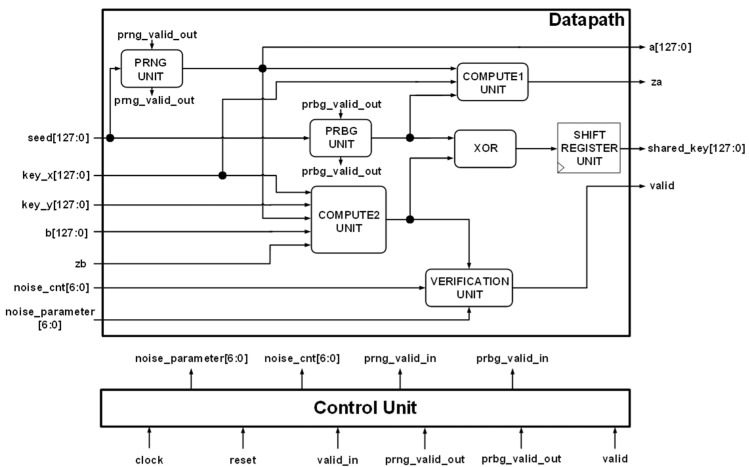
Proposed hardware architecture for the lightweight key exchange protocol.

**Figure 11 sensors-22-03004-f011:**
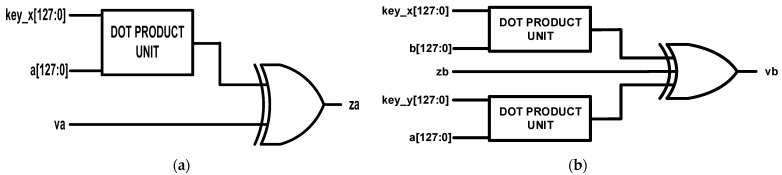
Proposed computation units: (**a**) COMPUT1 unit; (**b**) COMPUTE2 unit.

**Figure 12 sensors-22-03004-f012:**
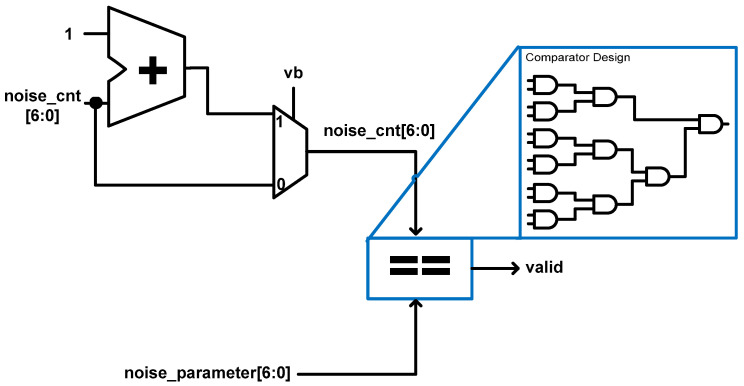
Proposed architecture for VERIFICATION unit.

**Figure 13 sensors-22-03004-f013:**
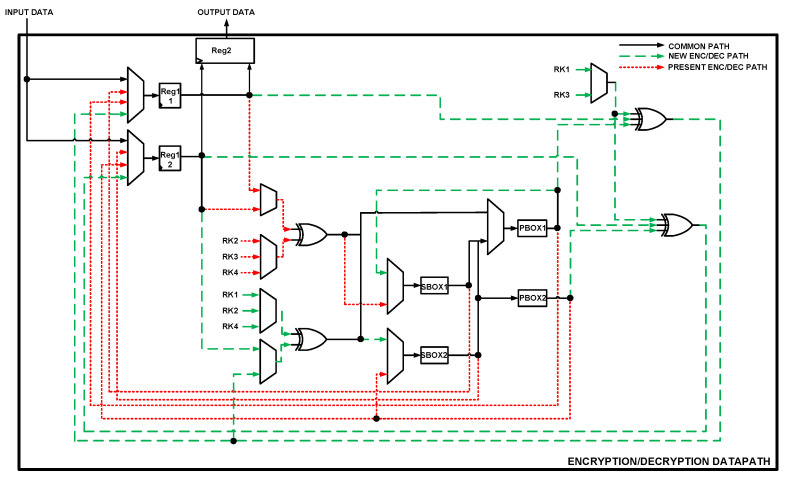
Proposed architecture for unifying two lightweight encryption/decryption protocols.

**Figure 14 sensors-22-03004-f014:**
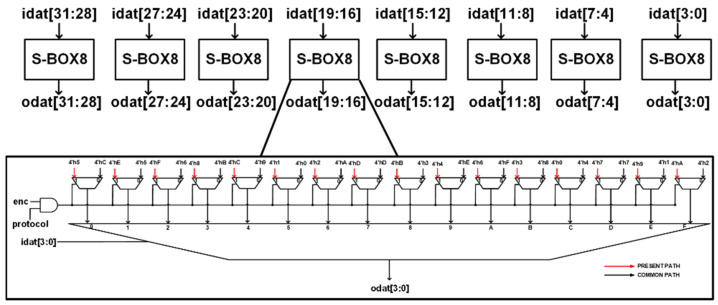
Proposed hardware architecture for 32-bit Unified S-BOX.

**Figure 15 sensors-22-03004-f015:**
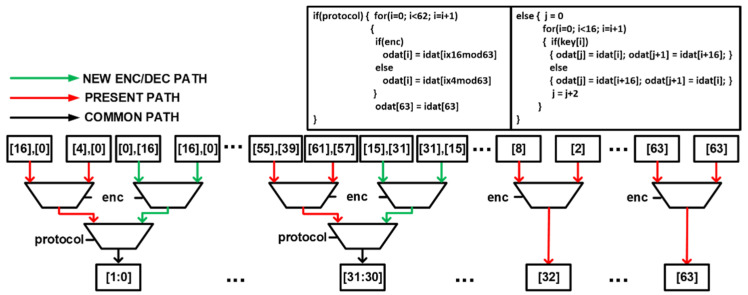
Proposed hardware architecture for 64-bit Unified P-BOX.

**Figure 16 sensors-22-03004-f016:**
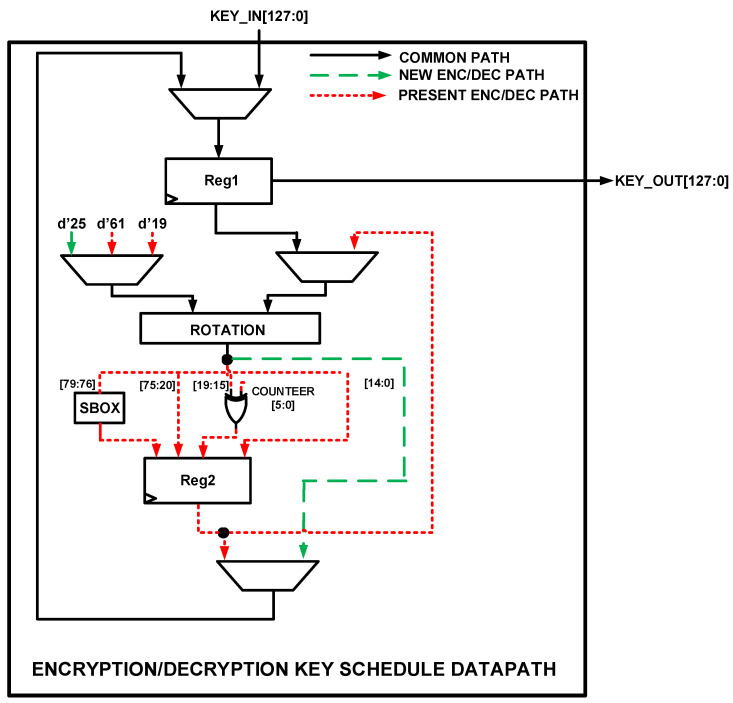
Proposed hardware architecture for unified key schedule.

**Figure 17 sensors-22-03004-f017:**
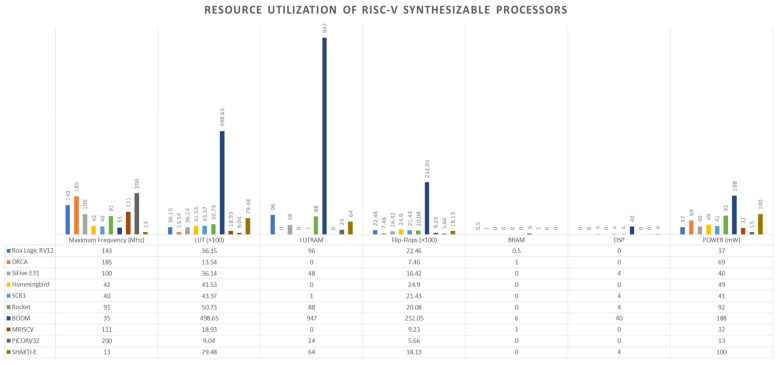
Synthesis results of RISC-V processor cores.

**Figure 18 sensors-22-03004-f018:**
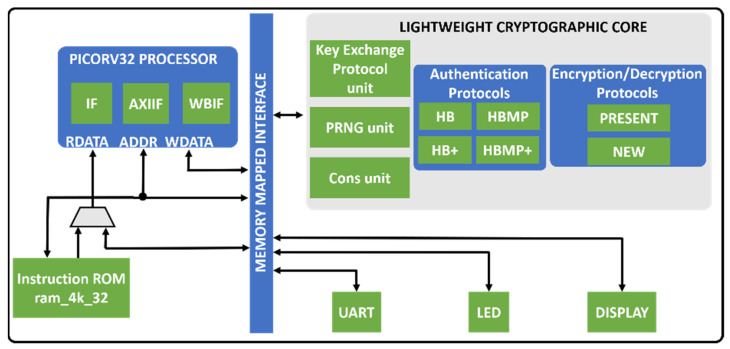
Overall hardware architecture for the proposed lightweight cryptographic SoC.

**Figure 19 sensors-22-03004-f019:**
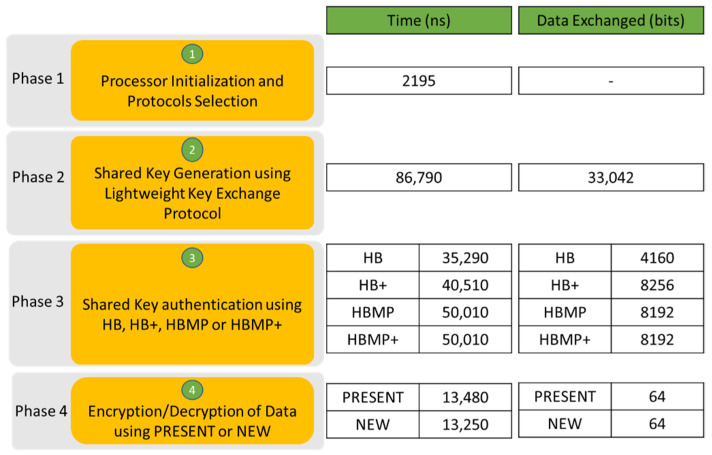
Simulation results.

**Figure 20 sensors-22-03004-f020:**
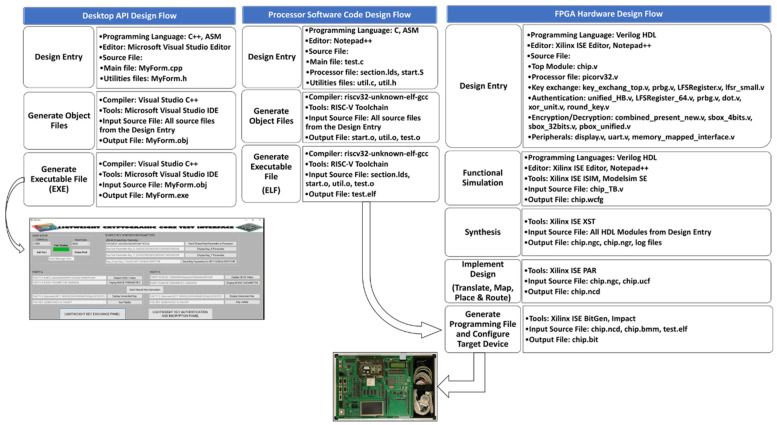
FPGA board test design flow.

**Figure 21 sensors-22-03004-f021:**
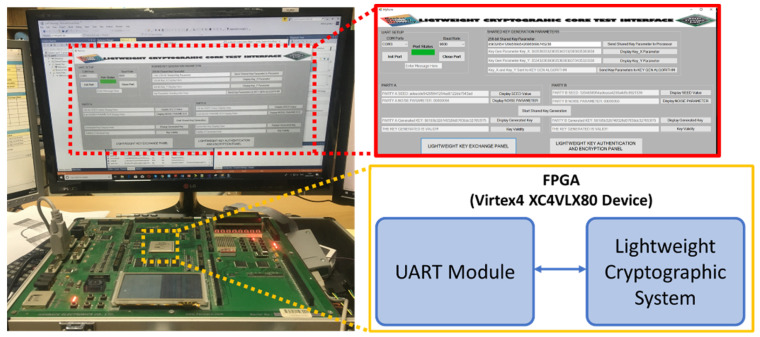
FPGA board setup for verifying the proposed lightweight cryptographic SoC.

**Figure 22 sensors-22-03004-f022:**
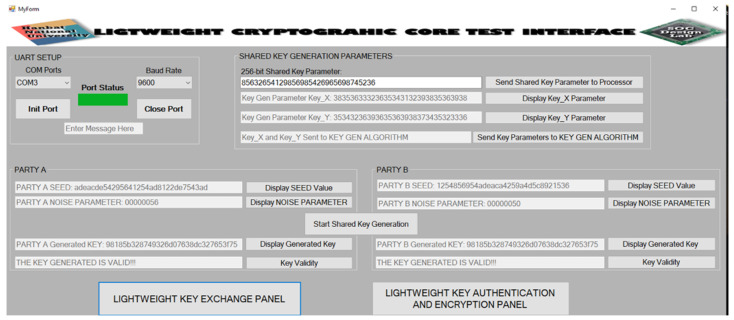
GUI of lightweight key exchange panel.

**Figure 23 sensors-22-03004-f023:**
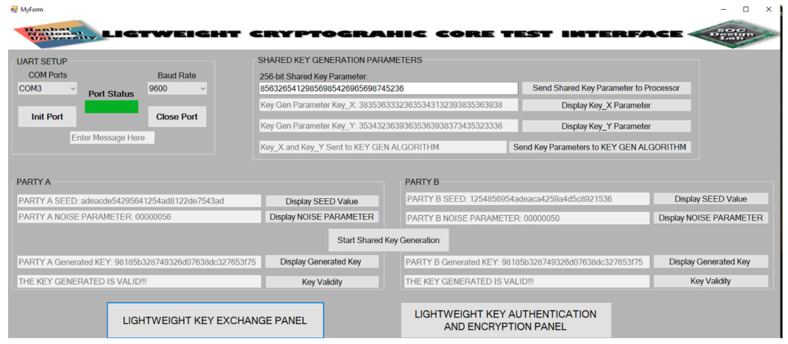
GUI of the lightweight key authentication and encryption/decryption panel.

**Table 1 sensors-22-03004-t001:** Specifications of EPC-C1G2 RFID tags.

Parameter	Limitation
Total Gate Count	1000–10,000 gates
Gate Count for Security Protocols	200–2000 gates
Reading Distance	3 m
Performance	100 reads/second
Clock Cycles per Read	10,000 Cycles
Power Consumption	10 mW
Bandwidth	200 Kbits/second
Cost	USD 0.1–USD 1

**Table 2 sensors-22-03004-t002:** Features of the lightweight authentication unit.

Feature	Proposed Design	Conventional Approach
Random Number (RN) Generation	Generates 64-bit RN by concatenating 4 16-bit Linear Feedback Shift Registers (LFSRs) Produces RN in 16 clock cycles Saves hardware area due to the use of flip-flops	Generates 64-bit RN by using AKARI (implemented with multipliers) Produces RN in 66 clock cycles Increase in hardware area due to the use of the multiplication operator
Dot Product	Computes the dot product of 64-bit values using 32 AND gates and 32 XOR gates Produces dot product in 1 clock cycle Increase in hardware area but a decrease in latency	Computes the dot product of 64-bit values using 1 AND gate and 1 XOR gate and a multiplexer Produces dot product in 64 clock cycles Reduction in the area but an increase in latency
Resource Sharing	The four protocols are combined into a single module to share resources The resource sharing leads to an area reduction of 28%	Most approaches design stand-alone architectures Non-resource sharing approach leads to an increase in area

**Table 3 sensors-22-03004-t003:** Features of the key exchange unit.

Feature	Proposed Design	Conventional Approach
Operators	Basic logic gates such as XOR This reduces area which is the basic requirement of low-cost devices	Exponentiation, division, inversion, multiplication This results in an increase in the area which is not suitable for low-cost devices
Random Number (RN) Generation	Generates 64-bit RN by concatenating 4 16-bit LFSRs Produces RN in 16 clock cycles Saves hardware area due to the use of flip-flops	Generates 64-bit RN by using AKARI (implemented with multipliers) Produces RN in 66 clock cycles Increase in hardware area due to the use of the multiplication operator
Dot Product	Computes the dot product of 64-bit values using 32 AND gates and 32 XOR gates Produces dot product in 1 clock cycle Increase in hardware area but a decrease in latency	Computes the dot product of 64-bit values using 1 AND gate and 1 XOR gate and a multiplexer Produces dot product in 64 clock cycles Reduction in the area but an increase in latency

**Table 4 sensors-22-03004-t004:** Features of the lightweight encryption/decryption unit.

Feature	Proposed Design	Conventional Approach
Substitution Box (S-Box)	The S-BOX and inverse S-BOX are combined into a 32-bit core This reduces the latency of the S-BOX operation This also reduces the output registers	S-BOX and inverse S-BOX designed separately using 4-bit input/output core This increases the latency of the S-BOX operation This also increases the output registers when the S-Boxes are implemented separately
Permutation Box (P-Box)	The P-BOX and inverse P-BOX of both ciphers are combined into a 64-bit core This reduces the output registers	The P-BOX and inverse P-BOX of both ciphers are implemented separately This increases the output registers
Key Schedule Algorithm	The key schedule algorithm of both ciphers is combined into a 128-bit core This reduces the output registers	The key schedule algorithm of both ciphers is implemented separately This increases the output registers
Resource Sharing	The two ciphers are combined into a single module to share resources The resource sharing leads to an area reduction of 5.9%	Most approaches design stand-alone architectures Non-resource sharing approach leads to an increase in area

**Table 5 sensors-22-03004-t005:** Features of the SoC-based lightweight cryptographic core.

Feature	Proposed Design	Conventional Approach
Algorithms	Lightweight authentication (HB, HB+, HBMP, HBMP+), lightweight key exchange, lightweight ciphers (PRESENT, NEW) Purely lightweight protocols that are suitable for low-cost devices	Compact versions of ECDSA, SHA-258, AES Compact versions of standard algorithms are not suitable for low-cost devices
Processor	RISC-V (PICORV32) Consume less than 10% of the gate counts	RISC-V (RV32I) Consume more than 20% of the gate count
Resource Sharing	The hardware architectures of the proposed design were made to share resources to reduce the area Results in 33 k gate count at 50 MHz frequency This is suitable for low-cost devices	All the hardware architectures were designed separately which leads to an increase in the area Results in 149 k gate count at 16 MHz frequency This is not suitable for low-cost devices

**Table 6 sensors-22-03004-t006:** Synthesis result for the proposed unified lightweight authentication architecture.

Design	Slices	LUTs	Frequency (MHz)
HB (Reader + Tag) [29]	96	110	-
HB+ (Reader + Tag) [29]	242	446	-
HBMP (Reader + Tag) [29]	396	700	-
HBMP+ (Reader + Tag) [29]	360	654	-
Total (HB, HB+, HBMP, HBMP+) [29]	1094	1910	-
Proposed Unified Architecture	786	1431	176.6

**Table 7 sensors-22-03004-t007:** Synthesis results for the proposed key exchange protocol hardware architecture.

Design	Area (Slices)	Frequency (MHz)	Latency (µs)	Power (mW)
[30]	12,881	100	31,930	-
[31]	9670	147	283	45
This Work	312	432	76	33

**Table 8 sensors-22-03004-t008:** Synthesis results for the proposed unified encryption/decryption hardware architecture.

Design	Slices	Flip-Flops	LUTs	Frequency (MHz)
PRESENT (ENCRYPTION + Decryption) [10]	304	402	530	-
NEW (ENCRYPTION + Decryption) [11]	530	392	990	-
Total (PRESENT [10] + NEW [11])	834	794	1520	-
Proposed Unified Architecture	863	199	1431	50

**Table 9 sensors-22-03004-t009:** Synthesis results for the proposed lightweight cryptographic SoC.

Design	Algorithms	Tech (nm)	Processor	Logic Gates	Frequency (MHz)	Efficiency
[19]	ECDSA SHA-258 AES-128	65	RISC-V RV32I	149 k	16	0.000107
This Work	Key Exchange HB, HB+, HBMP, HBMP+ PRESENT, NEW	130	RISC-V PICORV32	33 k	50	0.0015

## Data Availability

Not applicable.

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
