# Peer review of "A Lightweight System-On-Chip Based Cryptographic Core for Low-Cost Devices"

_sensors, 2022, doi:10.3390/s22083004_

Round 1

Reviewer 1 Report

The authors of this paper present an implementation of a System-on-Chip (SoC) cryptographic system including two encryption protocols, four authentication protocols, and one protocol to generate keys. The aim is to provide an alternative for secure low-cost devices.

One motivation for the authors' work is the focus on security for the Internet of Things. Thus, they provide an analysis of how much would it cost for users to get their devices hacked. Then, the aim is to merge four authentication (HB) protocols as well as the protocols mentioned above. Then, after surveying ten RISC-V processors, they select the most suited one for their proposal. Thereafter, to test the operation of their system an application is developed acting as an interface to the SoC hardware on an FPGA board. Moreover, a very important goal of the design developed is the final cost, so an interesting analysis is provided by developing a calculator to estimate the cost for users if their devices were used in DDoS attacks.

The system design is described by the authors in great detail, maybe too much detail in some parts of the paper. For example, the description started on pages 17-18 is perhaps extremely detailed regarding the goal of the paper. I am convinced the authors could be more concise in several parts of the paper since sometimes it would seem one was reading a hardware manual. Thus, saying so one may pose the question: Could a reader working in this field area be able to fairly reproduce the proposal? If not, it would be more appropriate to focus on the research contribution of the paper rather than on the engineering design. For example, one may wonder if some potential attacks could be made on the system proposed. If so, it would be interesting to report how it would behave under such attacks.

Reviewer 2 Report

  • There are several problems that minimize its overall contribution to the literature. What is your 'theory'? How has this paper contributed to existing theory? How has this paper advanced our understanding on the previous work? This is missing from your literature section. What is new or unexpected from this study
  • Explaining the problem and the gaps in existing literature in a concise but self-contained way (although readers might wish to consult references, they should not be forced to do so)
  • I suggest that the authors introduce certain taxonomy, at least through subsections.
  • The authors should clearly emphasize the contribution of the study. Please note that the up-to-date of references will contribute to the up-to-date of your manuscript.
  • The model parameter uncertainty are not mentioned. This study may be proposed for publication if it is addressed in the specified problems. The performance analysis of the system in terms of measurement uncertainty, real time working, and total measurement error should be provided.

  •   What is the computation time for the algorithm? Provide running time for the proposed method? Provide the comparison of computation time between the proposed method and other works.
  • Why the authors do obtained better results? What in their proposed architecture is the   key   ingredient   that   results   in   the   improved   results?
  • In introduction, the authors are requested to add a sub-section namely "need for research".
  • In Related work , the authors are requested to add a sub-section namely "need to extend the related work".
  • Limitations of the study should be mentioned in conclusion section.
  • The contribution is not clear, it is suggested to highlight the novelty of the paper after clearly introducing the main research gaps.
  • There are many algorithm parameters in the proposed method. What's the influence of these parameters?
  • motivating the design choices made, providing details about the alternatives

  • providing a clear yet mathematically rigorous description of the tools used

Reviewer 3 Report

What is scientific novelty in the presented approach? 
It is an engineering solution that implements known protocols and algorithms. The article needs to clearly define the scientific novelty and the scientific results obtained.
A review is not enough, the lack of review of related works. What are existing solutions and what are the unresolved issues? There are a lack of comparisons presented approach with existing solutions.

Round 2

Reviewer 1 Report

Thank you for the updated version of your article. The paper has been improved according to the suggested observations. On the other hand, you decided to leave for future work presenting some potential attacks on your system. While such a decision is acceptable given the nice work presented, I believe it would be great to add at least a couple of paragraphs describing how one potential attack on the system could be carried on.

Reviewer 2 Report

I think that all of my review' criticisms have been answered and the paper is now ready for acceptance.

Reviewer 3 Report

The authors corrected the article according to the comments.
